# Antioxidant Activity of Quercetin in a H_2_O_2_-Induced Oxidative Stress Model in Red Blood Cells: Functional Role of Band 3 Protein

**DOI:** 10.3390/ijms231910991

**Published:** 2022-09-20

**Authors:** Alessia Remigante, Sara Spinelli, Elisabetta Straface, Lucrezia Gambardella, Daniele Caruso, Giuseppe Falliti, Silvia Dossena, Angela Marino, Rossana Morabito

**Affiliations:** 1Department of Chemical, Biological, Pharmaceutical and Environmental Sciences, University of Messina, 98122 Messina, Italy; 2Institute of Pharmacology and Toxicology, Paracelsus Medical University, 5020 Salzburg, Austria; 3Biomarkers Unit, Center for Gender-Specific Medicine, Istituto Superiore di Sanità, 00161 Rome, Italy; 4Complex Operational Unit of Clinical Pathology, Papardo Hospital, 98122 Messina, Italy

**Keywords:** red blood cells, hydrogen peroxide, oxidative stress, quercetin, Band 3 protein, anion exchange

## Abstract

During their lifespan, red blood cells (RBCs) are exposed to a large number of stressors and are therefore considered as a suitable model to investigate cell response to oxidative stress (OS). This study was conducted to evaluate the potential beneficial effects of the natural antioxidant quercetin (Q) on an OS model represented by human RBCs treated with H_2_O_2_. Markers of OS, including % hemolysis, reactive oxygen species (ROS) production, thiobarbituric acid reactive substances (TBARS) levels, oxidation of protein sulfhydryl groups, CD47 and B3p expression, methemoglobin formation (% MetHb), as well as the anion exchange capability through Band 3 protein (B3p) have been analyzed in RBCs treated for 1 h with 20 mM H_2_O_2_ with or without pre-treatment for 1 h with 10 μM Q, or in RBCs pre-treated with 20 mM H_2_O_2_ and then exposed to 10 µM Q. The results show that pre-treatment with Q is more effective than post-treatment to counteract OS in RBCs. In particular, pre-exposure to Q avoided morphological alterations (formation of acanthocytes), prevented H_2_O_2_-induced OS damage, and restored the abnormal distribution of B3p and CD47 expression. Moreover, H_2_O_2_ exposure was associated with a decreased rate constant of SO_4_^2−^ uptake via B3p, as well as an increased MetHb formation. Both alterations have been attenuated by pre-treatment with 10 μM Q. These results contribute (1) to elucidate OS-related events in human RBCs, (2) propose Q as natural antioxidant to counteract OS-related alterations, and (3) identify B3p as a possible target for the treatment and prevention of OS-related disease conditions or aging-related complications impacting on RBCs physiology.

## 1. Introduction

Red blood cells (RBCs) are unique, highly specialized, and the most abundant cells in different organisms. Although their primary function is transportation of the respiratory gases O_2_ and CO_2_ between lungs and tissues, these circulatory cells are equipped with potent endogenous anti-oxidative systems that make them mobile free radical scavengers and provide antioxidant protection, not only to RBCs themselves but also to other tissues and organs in the body [1]. The production of reactive oxygen and nitrogen species (ROS/RNS) generated during cellular metabolism in biological systems is balanced by the ability of the latter to defend through their sophisticated antioxidant machinery [2,3,4,5,6]. Nevertheless, when oxidants are produced in excess, or when the antioxidant defenses regulating them are ineffective, this balance can be perturbed, thus resulting in oxidative stress (OS) [3,7,8,9,10,11]. In these conditions, organic biomolecules can be altered through oxidation to an extent that exceeds repair capacity [12]. Oxidative modifications lead to the accumulation of damaged proteins and oxidized lipids, which, when present in excess, impair cellular architecture and function, resulting in the life span of the cells being shortened [1,13]. Particularly, due to the high polyunsaturated fatty acids content in RBC membranes, high ROS levels can easily induce lipid peroxidation, an event that causes further damage in perturbing membrane stability and reducing RBC ability to resist to lysis [14].

The study of RBC morphology is of great importance in the field of hemorheology. The morphology of the circulating cells has a key influence on the rheological properties of the blood and changes in RBC morphology can lead to decreases in their deformability and increased aggregation [15,16,17]. These cells may respond to any form of insult by changing their morphology following changes in their membrane biochemical composition. During OS, biochemical changes include disruptions in the molecular arrangement of the bilayer, whereas biophysical changes are represented by alterations in the general structural arrangement (proteins and lipids) and erythrocyte morphology, which translate into changes in mechanical properties, such as deformability and aggregability [18,19,20,21,22]. These physicochemical adaptations are critically correlated with Band 3 protein (B3p) function [20,23,24,25,26,27].

Band 3 protein, also known as anion exchanger 1 (AE1), is encoded by the SLC4A1 gene and with more than 1 million copies per cell is the most abundant membrane protein in RBCs [28]. The crystal structure of B3p was recently obtained and revealed two domains, an N-terminal cytosolic domain that anchors the cytoskeleton at the plasma membrane and interacts with different proteins, and a C-terminal membrane domain that mediates the anion exchange [29,30]. ROS and/or RNS are inevitably produced during the process of oxygen delivery to tissues and induce oxidative damage to RBCs [31]. Since an imbalance in the physicochemical properties of RBCs can make them dysfunctional and impede an efficient tissue oxygenation, the maintenance of their functionality is of the outmost importance. The redox state in the RBC is normally regulated by a complex endogenous antioxidant system, which is composed of proteins with enzymatic activities, including glutathione peroxidase, catalase, and superoxide dismutase and non-enzymatic small-molecule compounds, such as glutathione, which are able to quickly neutralize ROS/RNS and ensure a low production of reactive species [1,17,32]. It should be pointed out that eventual oxidative damage goes beyond the cell itself. Abnormal quantities of reactive species leaving the damaged RBCs could harm other cell components in the bloodstream and tissues. To name just a few examples, ROS and RNS could be associated with endothelial cell aging and vascular damage, which lead to tissue hypoxia and promote inflammation and fibrosis [33,34]. In addition, hemoglobin denaturation and damage to the N-terminal of B3p are recognized as the starting events of immunological recognition and phagocytic removal of senescent and/or OS-impaired RBCs from blood circulation [23]. Thus, elucidating the pathways by which RBCs counteract OS can provide unique ways for cellular responses to oxidative damage.

Plant polyphenols are natural compounds showing potent antioxidant qualities that can help the activity of an endogenous antioxidant system, thus promoting the redox homoeostasis of the whole cell [35]. In particular, quercetin (3,5,7,3′,4′-pentahydroxyflavone; Q) is part of a group of bioflavonoids found in fruits and vegetables [36]. Quercetin is associated with various health benefits; in particular, its anti-oxidative capacity has been evaluated in healthy volunteers [37,38] and murine models [39,40,41]. This molecule can directly neutralize ROS/RNS and/or inactivate molecules with pro-oxidant capacity. It is well known that the stability of cell membranes can be positively affected by exogenous antioxidants [16,17] and several studies have showed a protective effect of Q on RBCs [42,43,44,45]. Because RBCs deliver oxygen to the entire body, the maintenance of a constant amount of RBCs is of pivotal importance for the health of an individual [46]. We hypothesized that Q could exert a beneficial role against oxidative injury in human RBCs. Thus, we explored the potential protective effect of Q (10 µM) in a model of OS represented by human RBCs treated with non-toxic concentrations of H_2_O_2_ [47]. This cell-based model could represent those human pathologic conditions having acute OS as hallmark, including hemolytic anemia, vaso-occlusion, and progressive vascular injury affecting multiple organ systems, thus affecting RBC integrity. Since we have demonstrated that B3p function as a sensitive tool to assess the impact of OS on homeostasis of RBCs [48,49,50,51,52,53,54], the anion exchange capability through B3p and the effects of OS on cell membrane structure have been evaluated.

## 2. Results

### 2.1. Measurement of Percentage Hemolysis

Figure 1 reports the hemolysis measurement in human RBCs showed as percentage of hemoglobin released in a 0.9% NaCl solution at room temperature (25 °C). Red blood cells were treated for 1 h at 25 °C with 20 mM H_2_O_2_. This exposure did not increase hemolysis percentage compared to cells left untreated (control). Quercetin alone (1 h at 37 °C) did not significantly affect hemolysis. Based on this result, 20 mM H_2_O_2_ concentration has been used for the experiments shown in the following.

### 2.2. Evaluation of Erythrocyte Cell Shape

In the following, the experimental design shown in Figure 2 applies.

As depicted in Figure 3, incubation for 1 h at 25 °C with 20 mM H_2_O_2_ induced morphological alterations of RBCs. In fact, in this condition we detected 37.1% of acanthocytes (RBCs with surface blebs) by scanning electron microscopy analysis. However, in samples pre-treated with 10 μM Q and then treated with 20 mM H_2_O_2_, the percentage of morphologically altered cells was reduced to 6.5%. Instead, in RBCs treated with 20 mM H_2_O_2_ and then exposed to 10 μM Q we still detected 18.4% of acanthocytes. Therefore, the formation of acanthocytes was not completely avoided by Q post-treatment (Table 1).

### 2.3. Oxidative Stress Assessment

#### 2.3.1. Evaluation of Intracellular ROS Levels

The evaluation of ROS levels was carried out by flow cytometry in RBCs left untreated or, alternatively, exposed to H_2_O_2_ with or without pre-exposure to 10 µM Q for 1 h or pre-treated with 20 mM H_2_O_2_ and then exposed to 10 µM Q. Figure 4A shows the intracellular ROS levels at different time points (0, 15, 30, 45, 60 min) after incubation with H_2_O_2_. Samples exposed to 20 mM H_2_O_2_ showed a significant increase of ROS levels compared to the untreated samples. After a 30 min treatment, levels of ROS increased by 50% in H_2_O_2_-treated samples and remained unchanged over time. In Figure 4A, the effect of Q is also reported. In samples pre- or post-exposed to 10 µM Q, 20 mM H_2_O_2_ failed to significantly increase ROS levels, which remained unchanged when compared to control values.

#### 2.3.2. Measurement of Thiobarbituric Acid Reactive Substances (TBARS) Levels

Thiobarbituric acid reactive substances (TBARS) measurement in RBCs is reported in Figure 4B. As expected, TBARS levels of RBCs treated with 20 mM H_2_O_2_ for 1 h were significantly higher than those of RBCs left untreated (control). Importantly, in RBCs pre-treated with 10 µM Q and exposed to 20 mM H_2_O_2_ or pre-treated with 20 mM H_2_O_2_ and then exposed to 10 µM Q, TBARS levels were significantly reduced compared to those measured in 20 mM H_2_O_2_-treated RBCs. Of note, Q alone did not significantly affect TBARS levels.

#### 2.3.3. Total Sulfhydryl Group Content Measurement

Figure 4C shows the total content of sulfhydryl groups (µM TNB/µg protein) in RBCs left untreated or treated with either the oxidizing compound NEM (2 mM for 1 h at 25 °C, as the positive control), or 10 µM Q for 1 h, or 20 mM H_2_O_2_ for 1 h with or without pre-treatment with 10 µM Q, or pre-treated with 20 mM H_2_O_2_ and then exposed to 10 µM Q. As expected, exposure to NEM led to a significant reduction in sulfhydryl group content. Sulfhydryl groups in 20 mM H_2_O_2_-treated RBCs were also significantly reduced in respect to control (left untreated). Pre- or post-treatment with 10 µM Q significantly restored the total levels of sulfhydryl groups in 20 mM H_2_O_2_-treated RBCs. Quercetin alone did not significantly affect total sulfhydryl group content.

#### 2.3.4. Evaluation of Methemoglobin (MetHb) Levels

Figure 4D shows MetHb levels (% MetHb) measured in RBCs left untreated or treated with 20 mM H_2_O_2_ (1 h at 25 °C) with or without pre-treatment with 10 µM Q (1 h at 37 °C), or pre-treated with 20 mM H_2_O_2_ and then exposed to 10 µM Q, or alternatively, treated with the well-known MetHb-forming agent NaNO_2_ (2 mM for 1 h at 25 °C). The levels of MetHb measured after incubation with NaNO_2_ were significantly higher than those detected in RBCs left untreated (control). In parallel, MetHb levels measured following exposure to 20 mM H_2_O_2_ were significantly higher than those measured in control (left untreated). Pre-exposure to 10 µM Q significantly reduced the MetHb levels in H_2_O_2_-treated RBCs towards values that did not differ from control values. On the contrary, MetHb levels measured in RBCs pre-treated with 20 mM H_2_O_2_ and then exposed to 10 µM Q were significantly increased with respect to those of RBCs left untreated (control), but did not differ with respect to those of H_2_O_2_-treated RBCs. Quercetin alone did not significantly affect the % MetHb.

### 2.4. CD47 Expression Level Determination

Flow cytometry analysis has shown a significantly decreased expression of CD47 in RBCs treated with 20 mM H_2_O_2_ for 1 h compared to untreated (control) samples (Figure 5A). Conversely, the expression of this protein was significantly restored by both pre- and post-treatment with 10 µM Q in H_2_O_2_-treated RBCs. Quercetin alone did not significantly affect CD47 expression. Data obtained by flow cytometry were confirmed by immunofluorescence analyses, which reported a remarkable rearrangement and redistribution of this protein (Figure 5B).

### 2.5. Band 3 Protein Expression Level Determination

B3p protein expression was found significantly decreased in human RBCs treated with 20 mM H_2_O_2_ for 1 h with respect to those left untreated (control) (Figure 6A). Band 3 protein expression was significantly restored in RBCs both pre- and post-treated with 10 µM Q (Figure 6A). Quercetin alone did not significantly affect B3p expression. In addition, a redistribution of B3p was detected by immunofluorescence. Specifically, B3p was mainly localized in blebs of acanthocytes after treatment with 20 mM H_2_O_2_, with respect to untreated RBCs (Figure 6B). These changes were attenuated by Q.

### 2.6. Measurement of SO_4_^2−^ Uptake via B3p

Figure 7 reports the SO_4_^2−^ uptake as a function of time in RBCs left untreated (control) and in RBCs treated with 20 mM H_2_O_2_ (1 h at 25 °C) with or without Q (pre- or post- treatment, 1 h at 37 °C)_._ In control conditions, SO_4_^2−^ uptake progressively increased and reached equilibrium within 45 min (rate constant of SO_4_^2−^ uptake = 0.056 ± 0.001 min^−1^). Red blood cells treated with 10 µM Q alone (1 h at 37 °C) showed a rate constant of SO_4_^2−^ uptake that was not significantly different with respect to control. Vice versa, the rate constant value (0.042 ± 0.001 min^−1^) in RBCs treated with 20 mM H_2_O_2_ was significantly decreased with respect to control (*** *p* < 0.001). In RBCs pre-incubated with 10 µM Q (1 h at 37 °C) and then treated with 20 mM H_2_O_2_ (1 h at 25 °C), the rate constant (0.062 ± 0.001 min^−1^) was significantly higher than that of RBCs treated with 20 mM H_2_O_2_ (0.042 ± 0.001 min^−1^) and was not significantly different with respect to control (Table 2). Similarly, in RBCs incubated with 20 mM H_2_O_2_ (1 h at 25 °C) and then exposed to 10 µM Q (1 h at 37 °C), the rate constant (0.047 ± 0.001 min^−1^) was significantly different than that of RBCs treated with 20 mM H_2_O_2_ (0.042 ± 0.001 min^−1^) (Table 2). SO_4_^2−^ uptake was almost completely blocked by 10 µM DIDS applied at the beginning of incubation in SO_4_^2−^ medium (0.017 ± 0.001 min^−1^, *** *p* < 0.001, Table 2). Additionally, the SO_4_^2−^ amount internalized by H_2_O_2_-treated RBCs after 45 min of incubation in SO_4_^2−^ medium was significantly lower compared to control (Table 2), while it was not significantly different in RBCs pre-incubated with 10 µM Q and then exposed to 20 mM H_2_O_2_ (Table 2). Conversely, the SO_4_^2−^ amount internalized by RBCs incubated with 20 mM H_2_O_2_ and then treated with 10 µM Q was significantly lower compared to those left untreated (** *p* < 0.01, Table 1) and was not significantly different with respect to 20 mM H_2_O_2_. In DIDS-treated cells, the SO_4_^2−^ amount internalized (5.49 ± 2.50) was significantly lower than that determined in both control and treated RBCs (*** *p* < 0.001, Table 2).

## 3. Discussion

Red blood cells are mainly known as transporters of metabolic gases and nutrients for tissues and perform additional important biological functions, such as the regulation of redox balance. Moreover, RBCs are deeply sensitive to external stressors and important health indicators [55]. During increase of OS levels, RBCs are exposed to circulating oxidative agents and their redox homeostasis is compromised due to the disequilibrium between pro-oxidant and anti-oxidant species, thus leading to a variety of structural alterations that promptly signal a physiological derangement [1,56]. In the last period, increasing evidence focused on beneficial effects of natural antioxidants and their ability to counteract OS-induced pathological conditions. In fact, dietary supplementation with fruits and vegetables, rich in phytochemicals, has been found to provide several health benefits. Quercetin is a naturally available flavonoid that showed different beneficial activities, including anti-oxidant properties, in different experimental models and therefore represents a good food supplement candidate [39,41,43,57,58].

In this investigation, the ameliorative effect of Q on H_2_O_2_-induced OS in RBCs was assessed. The results shown here indicate that Q could improve both structure and function of RBCs as well as potential pathological events that are closely related to oxidative damage. There are many studies describing the multiple activities of flavonoids, however, the effects of Q on human RBCs following H_2_O_2_ treatment (20 mM) have not been evaluated. Firstly, this study tested the H_2_O_2_ concentration to exclude a possible hemolytic power. As shown in Figure 1, 20 mM H_2_O_2_ did not cause hemolytic events after 1 h incubation compared to control. Based on this finding, 20 mM H_2_O_2_ was used to model oxidant conditions in RBCs.

Susceptibility of RBCs to H_2_O_2_ exposure was studied in terms of morphological alterations by SEM. The images (Figure 3) show changes in the RBC shape, with the typical biconcave shape lost in a remarkable number of cells, which shows surface blebs, namely acanthocytes. However, pre-treatment with Q attenuated the morphological changes, with a reduction of acanthocytes numbers. Conversely, treatment with Q post- H_2_O_2_ still revealed a cell morphology alteration, pointing to irreversible mechanisms underlying the changes in erythrocyte shape (Table 1). The structure of the circulating cells has a crucial influence on the rheological properties of the blood and changes in morphology can lead to decreased deformability and increased aggregation [15,16,22]. To better clarify the mechanisms underlying these structural changes, some parameters related to OS assessment were monitored. Red blood cells were able to respond to any form of insult by changing their morphology following changes in their membrane or biochemical composition. Different phenomena, including oxidation of sulfhydryl groups of membrane proteins, oxidation of membrane fatty acid residues, or oxidation of hemoglobin, could alter membrane properties and cell shape, thus causing loss of membrane integrity and decreased deformability. Since oxidation of biological macromolecules, such as lipids and proteins, derives from the deleterious effects of ROS generated during cellular metabolism, intracellular ROS levels have been evaluated. Our findings showed that treatment with Q both pre- and post-exposure to 20 mM H_2_O_2_ induced a decrease of ROS levels (Figure 4A). This finding is supported by other authors and suggests that dietary supplementation with flavonoids has an effective anti-oxidant activity, which is generally attributed to their ability to directly scavenge ROS [59].

Due to oxidation stress, polyunsaturated fatty acids of RBC membranes are damaged. This results in a steep increase in malondialdehyde (MDA) and thiobarbituric acid (TBA), which are biomarkers currently used to reveal oxidation of lipids under different conditions. Our results showed that exposure to 10 µM Q before or after OS avoided the lipid peroxidation of membranes induced by treatment with 20 mM H_2_O_2_ (Figure 4B). However, the ROS attack on lipids initiates a chain reaction, which leads to generation of more ROS that can harm other cellular components, including proteins. In this regard, RBCs represent a convenient model not only to investigate the degree of lipid peroxidation, but also oxidative damage on a protein level, being the protein component particularly abundant in these peculiar cells. Therefore, the sulfhydryl group content of total proteins was also evaluated. Exposure to Q before or after OS protected erythrocyte proteins from oxidative damage (Figure 4C), which is in line with what has been previously demonstrated by other authors [60,61].

Elevation of OS has been associated with eryptosis. By analogy with apoptosis of nucleated cells, RBCs may undergo programmed cell death, which is characterized by cell shrinkage and cell membrane phospholipid scrambling. The eryptosis machinery includes activation of redox-sensitive Ca^2+^-permeable cation channels resulting in Ca^2+^ entry. Cytosolic Ca^2+^ elevation further activates erythrocyte scramblase and calpain resulting in phosphatidylserine (PS) externalization and membrane blebbing, respectively [62,63]. Phosphatidylserine-exposing RBCs are rapidly phagocytosed and, thus, cleared from circulating blood. However, no externalization of PS was found in RBCs treated with 20 mM H_2_O_2_, thus denoting that, in this model, RBCs do not enter apoptosis but remain in an early phase of the oxidation process, which is critically important to permit the action of antioxidants. The importance of this time window in the prevention of OS-related alterations has been very recently confirmed by our research group in a model of RBCs aging based on D-Galactose exposure. In this model, where PS scrambling did not occur OS-related derangements could be reversed by pre-treatment with an Açai berry extract, which is particularly rich in flavonoids [64].

Another major feature of RBC oxidation is the clustering and/or the breakdown of B3p and the binding of oxidized hemoglobin (MetHb) to high affinity sites on B3p. To better investigate the role of B3p, the expression of this protein during OS was also investigated. In particular, data showed that, following 20 mM H_2_O_2_ treatment, B3p re-arranged in surface blebs (Figure 6C). The N-terminal cytoplasmic domain of B3p contains binding sites for cytoskeletal and cytoplasmic proteins, including hemoglobin [65,66,67]. In RBCs, ROS production led to hemoglobin denaturation and further release of heme iron. This process can be autocatalytic, leading to an ever-increasing OS once it is initiated by the release of threshold amounts of free iron. Not surprisingly, healthy RBCs are equipped with multiple mechanisms to inactivate potent oxidants (e.g., MetHb). To better elucidate the molecular interaction between B3p and oxidized hemoglobin, MetHb levels were also evaluated. Our data indicated that exposure to 20 mM H_2_O_2_ for 1 h increased the levels of MetHb in RBCs (Figure 4D). These modifications can start a cascade of biochemical and structural transformations, including the release of microparticles containing hemicromes and clustering of B3p regions, as previously demonstrated by other groups [23,24,68]. When the oxidation processes are advanced, these clusters could provide a recognition site for antibodies directed against aging cells, thus triggering the premature removal of senescent RBCs from the circulation at the end of their 120-day life span. Interestingly, only pre-treatment with Q prevented H_2_O_2_-induced MetHb formation (Figure 4C), which was still observed when Q was applied after OS.

One of the most interesting and still poorly investigated implications of OS is its impact on membrane transport systems. To define the possible effects of 20 mM H_2_O_2_ on RBC functional activity, SO_4_^2−^ uptake was measured by means of a validated method to assay anion exchange capability through B3p. In RBCs treated with 20 mM H_2_O_2_, the rate constant for SO_4_^2−^ uptake was decreased compared to control (Figure 7) and, in parallel, the amount of internalized SO_4_^2−^ was significantly reduced (Table 2). However, 1 h pre-treatment with 10 µM Q completely restored the rate constant of SO_4_^2−^ uptake, as well as the amount of internalized SO_4_^2−^, thus demonstrating a beneficial effect of Q on B3p function. On the contrary, Q application after exposure to H_2_O_2_ restored neither the rate constant of SO_4_^2−^ uptake (Figure 7) nor the amount of SO_4_^2−^ internalized (Table 2). In this regard, we suggest that the mechanisms by which high levels of H_2_O_2_, or eventually other oxidants, induce changes in the RBC function are due to reactions taking place in the cell interior and involving oxidation of heme proteins, which may result in cross-linking with the cytoplasmic domain of B3p.

This, and our former work, reported that B3p exhibits changes in the rate constant for SO_4_^2−^ uptake after exposure of RBCs to OS. In particular, non-hemolytic concentrations of H_2_O_2_ induced OS and provoked a decrease in the rate constant for SO_4_^2−^ uptake through B3p [53,54]. However, melatonin pre-treatment ameliorated the reduction in the rate constant for SO_4_^2−^ uptake, as well as the reduction in B3p expression that was observed following treatment with H_2_O_2_ [50]. Furthermore, the reduction of B3p anion exchange efficiency caused by a mild OS was prevented or attenuated by a short-time pre-incubation of RBCs with low H_2_O_2_ doses. This pre-incubation encourages RBCs to adapt to a mild and transient OS and favors an increased tolerance to a successive stronger oxidant condition. Such adaptation response, termed pre-conditioning, was monitored by measuring B3p activity, did not involve B3p-related Tyr-phosphorylation pathways, but was mediated by an increased activity of catalase [52]. In summary, a reduction of the transport rate following H_2_O_2_ exposure is most likely linked to the formation of oxidizing hemoglobin following OS. Therefore, the different effect on the SO_4_^2−^ transport kinetics observed in a former study is not surprising. However, in different models of OS obtained by exposing RBCs to high concentrations of D-Glucose or D-Galactose, we measured an acceleration of the anion exchange through B3p instead of a reduction. It is tempting to speculate that such a two-sided effect on anion exchange velocity depends on the specific structure targeted by the stressors and on the possible underlying pathways [49,69,70].

At last, our focus has been addressed to assay CD47 protein expression, which is a specific marker of “self” [71]. In fact, expression of CD47 on RBCs’ surface could be one of the mechanisms that regulate the removal of senescent cells from the bloodstream by phagocytosis. Treatment with 20 mM H_2_O_2_ for 1 h did not induce a remarkable loss of CD47 protein, but rather its rearrangement and redistribution (Figure 5C), which could be linked to the progressive plasma membrane blebbing and vesiculation, respectively (Figure 3). In this model, CD47 expression and distribution were completely re-established in RBCs pre-treated with Q as well as RBCs treated with Q after exposure to H_2_O_2_. During their lifetime, RBCs release plasma membrane-derived ectosomes. These structures are generated by outward budding of the plasma membrane, followed by vesicle shedding [72,73]. Vesiculation is a way to remove dangerous molecules, such as oxidized proteins or denatured hemoglobin. Changes in lipids and proteins of RBC membranes induce a decrease in erythrocyte deformability and unfavorable changes in blood flow, which promote additional OS, proneness to atherosclerotic lesions, and increase in blood viscosity. As a result of ectosome formation, the protein composition of RBCs varies among circulating RBCs, with new RBCs being larger with a full equipment of membrane proteins and old RBCs smaller and denser with significantly lower content in membrane proteins, including CD47 and B3p [74]. In our model, the vesiculation process is missing, thus demonstrating that RBCs remain in an acute and early phase of the oxidation process.

## 4. Materials and Methods

### 4.1. Solutions and Chemicals

All chemicals were purchased from Sigma (Sigma, Milan, Italy). Both 3,3,4,5,7-pentahydroxyflavone (Q) and 4,4′-diisothiocyanatostilbene-2,2′-disulfonate (DIDS) stock solutions (10 mM) were prepared in dimethyl sulfoxide (DMSO). Quercetin (CAS Number: 117-39-5) was kindly provided by Professor Marika Cordaro from University of Messina. N-ethylmaleimide (NEM) stock solution (310 mM) was prepared in ethanol. H_2_O_2_ was diluted in distilled water from a 30% *w*/*w* stock solution. Both ethanol and DMSO never exceeded 0.001% *v*/*v* in the experimental solutions and were previously tested on RBCs to exclude hemolysis.

### 4.2. Erythrocyte Preparation

This study was prospectively reviewed and approved by a duly constitute Ethics Committee (prot.52-22, 20 April 2022). Upon informed consent, whole human blood samples from healthy volunteers were collected in test tubes containing ethylenediaminetetraacetic acid (EDTA). Plasma concentration of glycated hemoglobin (A1c) was less than 5%. Red blood cells were washed in isotonic solution (NaCl 150, 4-(2-hydroxyethyl)-1-piperazineethanesulfonic acid (HEPES) 5, Glucose 5, pH 7.4, osmotic pressure 300 mOsm/kgH_2_O) and centrifuged thrice (Neya 16R, 1200× *g*, 5 min) to delete plasma and buffy coat, respectively. Red blood cells were then suspended at specific hematocrits in isotonic solution and addressed to downstream analysis.

### 4.3. Percentage Hemolysis Measurement

To verify the % hemolysis, RBCs (35% hematocrit) were treated with 20 mM H_2_O_2_ (1 h at 25 °C) or 10 µM Q (1 h at 37 °C) in isotonic solution, suspended at 0.5% hematocrit in isotonic solution, centrifuged (Neya 16R, 1200× *g*, 5 min), and resuspended at 0.05% hematocrit in a 0.9% *v*/*v* NaCl solution [49,75]. Absorbance of hemoglobin was measured at 405 nm wavelength and subtracted for the absorbance of blank (0.9% *v*/*v* NaCl solution).

### 4.4. Analysis of Cell Shape by Scanning Electron Microscopy (SEM)

Samples, which were left untreated or exposed to 20 mM H_2_O_2_ (1 h at 25 °C) with or without 10 µM Q (1 h at 37 °C, before or after treatment with H_2_O_2_) were collected, plated on poly-l-lysine-coated slides, and fixed with 2.5% glutaraldehyde in 0.1 M cacodylate buffer (pH 7.4) at 25 °C for 20 min. Then, samples were post-fixed with 1% OsO_4_ in 0.1 M sodium cacodylate buffer and dehydrated through a graded series of ethanol solutions (from 30% to 100%). Absolute ethanol was gradually substituted by a 1:1 solution of hexamethyldisilazane (HMDS)/absolute ethanol and successively by pure HMDS. Afterwards, HMDS was completely removed, and samples were dried in a desiccator. Dried samples were mounted on stubs, coated with gold (10 nm), and analyzed by a Cambridge 360 scanning electron microscope (Leica Microsystem, Wetzlar, Germany) [76]. Altered erythrocyte shape was evaluated by counting ≥500 cells (50 RBCs for each different scanning electron microscopy (SEM) field at a magnification of 3000×) from samples in triplicate.

### 4.5. Measurement of Oxidative Stress Assessment

#### 4.5.1. Detection of Reactive Oxygen Species (ROS)

To evaluate intracellular reactive oxygen species, red blood cells, which were left untreated or treated with 20 mM H_2_O_2_ (1 h at 25 °C) with or without 10 µM Q (1 h at 37 °C before or after treatment with H_2_O_2_), were incubated in Hanks’ balanced salt solution, pH 7.4, containing dihydrorhodamine 123 (DHR 123; Molecular Probes) and then analyzed with a FACScan flow cytometer (Becton Dickinson, Mountain View, CA, USA). At least 20,000 events were acquired. The median values of fluorescence intensity were used to provide a semi-quantitative analysis of ROS production [77].

#### 4.5.2. Thiobarbituric-Acid-Reactive Substances (TBARS) Level Determination

TBARS levels were measured as described by Mendanha and collaborators [78], with slight modifications. Red blood cells were suspended at 20% hematocrit and incubated with 20 mM H_2_O_2_ (1 h at 25 °C) with or without 10 µM Q (1 h at 37 °C before or after treatment with H_2_O_2_). Afterwards, samples were centrifuged (Neya 16R, 1200× *g*, 5 min) and suspended in isotonic solution. Red blood cells (1.5 mL) were treated with 10% (*w*/*v*) trichloroacetic acid (TCA) and centrifuged (Neya 16R, 3000× *g*, 10 min). TBA (1% in hot distilled water, 1 mL) was added to the supernatant and the mixture was incubated at 95 °C for 30 min. Finally, TBARS levels were obtained by subtracting 20% of the absorbance at 453 nm from the absorbance at 532 nm (Onda Spectrophotometer, UV-21). Results are indicated as µM TBARS levels (1.56 × 10^5^ M^−^^1^ cm^−^^1^ molar extinction coefficient).

#### 4.5.3. Total Sulfhydryl Group Content Determination

Measurement of total -SH groups was carried out according to the method of Aksenov and Markesbery [79], with some modifications. Red blood cells (35% hematocrit), left untreated or treated with to 20 mM H_2_O_2_ (1 h at 25 °C) with or without 10 µM Q (1 h at 37 °C before or after treatment with H_2_O_2_), were centrifuged (Neya 16R, 1200× *g*, 5 min) and a sample of 100 µL was hemolyzed in 1 mL of distilled water. A 50 μL aliquot was added to 1 mL of phosphate-buffered saline (PBS, pH 7.4) containing EDTA (1 mM). 5,5′-Dithiobis (2-nitrobenzoic acid) (DTNB, 10 mM, 30 μL) was added to start the reaction and the samples were incubated for 30 min at 25 °C protected from light. Control samples, without cell lysate or DTNB, were processed concurrently. After incubation, sample absorbance was measured at 412 nm (Onda spectrophotometer, UV-21) and 3-thio-2-nitro-benzoic acid (TNB) levels were detected after subtraction of blank absorbance (samples containing only DTNB). To achieve a complete oxidation of -SH groups, an aliquot of red blood cells (positive control) was incubated with 2 mM NEM for 1 h at 25 °C [53,80]. Obtained values were normalized to protein content and results reported as μM TNB/mg protein.

#### 4.5.4. Measurement of Methemoglobin (MetHb) Levels

MetHb content was determined as reported by Naoum and collaborators with some modifications. This assay is based on MetHb and (oxy)-hemoglobin (Hb) determination by spectrophotometry at 630 and 540 nm wavelength, respectively. After incubation (20 mM H_2_O_2_ for 1 h at 25 °C with or without pre- or post-treatment with 10 µM Q for 1 h at 37 °C), 25 μL of red blood cells at 40% hematocrit were lysed in 1975 μL hypotonic buffer (2.5 mM NaH_2_PO_4_, pH 7.4; 4 °C). Then, samples were centrifuged (13,000× *g*, 15 min, 4 °C; Eppendorf) to remove membranes. The absorbance of the supernatant was measured (BioPhotometer Plus; Eppendorf). Incubation with 4 mM NaNO_2_ (for 1 h at 25 °C), a well-known MetHb-forming agent, was used to obtain a complete Hb oxidation [81]. The percentage (%) of MetHb was determined as follows: % MetHb = (OD630/OD540) × 100 (OD is optical density).

### 4.6. Analysis of Analytical Cytology

Red blood cells, which were left untreated or exposed to 20 mM H_2_O_2_ (1 h at 25 °C) with or without 10 µM Q (1 h at 37 °C before or after treatment with H_2_O_2_), were fixed with 3.7% formaldehyde in PBS (pH 7.4) for 10 min at 25 °C and then washed in the same buffer. Red blood cells were then permeabilized with 0.5% Triton X-100 in PBS for 5 min at 25 °C. After washing with PBS, samples were incubated with monoclonal anti-Band 3 protein (Sigma, St. Louis, MI, USA) or monoclonal anti-CD47 (Santa Cruz Biotechnology, Dallas, TX, USA) antibodies for 30 min at 37 °C, washed, and then incubated with a fluorescein isothiocyanate (FITC)-labeled anti-mouse antibody (Sigma) for 30 min at 37 °C [33]. Cells incubated with the secondary antibody given alone were used as negative control. Samples were analyzed by an Olympus BX51 Microphot fluorescence microscope or by a FACScan flow cytometer (Becton Dickinson, Franklin Lakes, NJ, USA) equipped with a 488 nm argon laser. At least 20,000 events have been acquired. The median values of fluorescence intensity were used to provide a semiquantitative analysis. Fluorescence intensity values were normalized for those of untreated RBCs and expressed in %.

### 4.7. SO_4_^2−^ Uptake Measurement

#### 4.7.1. Control Condition

SO_4_^2−^ uptake measurement was used to evaluate the anion exchange through B3p, as described elsewhere [51,82,83,84]. In short, after washing, red blood cells were suspended to 3% hematocrit in 35 mL SO_4_^2−^ medium (Na_2_SO_4_ 118, HEPES 10, glucose 5, pH 7.4, osmotic pressure 300 mOsm/kgH_2_O) and incubated at 25 °C in this medium. After 5, 10, 15, 30, 45, 60, 90, and 120 min, DIDS (10 μM), which is an inhibitor of B3p activity [85,86], was added to 5 mL sample aliquots, which were kept on ice. Afterwards, cells were washed three times in cold isotonic solution and centrifuged (Neya 16R, 4 °C, 1200× *g*, 5 min) to remove SO_4_^2−^ from the external medium. Distilled water (1 mL) was added to induce osmotic lysis of cells and perchloric acid (4% *v*/*v*) was used to precipitate proteins. After centrifugation (Neya 16R, 4 °C, 2500× *g*, 10 min), the supernatant containing SO_4_^2−^ trapped by cells was directed to the turbidimetric analysis. Supernatant (500 μL from each sample) was sequentially mixed to 500 μL glycerol diluted (1:1) in distilled water, 1 mL 4 M NaCl, and 500 μL 1.24 M BaCl_2_•2H_2_O. At last, the absorbance of each sample was measured at 425 nm (Onda Spectrophotometer, UV-21). The absorbance was converted to [SO_4_^2−^] L cells × 10^−2^ by means of a standard curve previously obtained by precipitating known SO_4_^2−^ concentrations. The rate constant of SO_4_^2−^ uptake (min^−^^1^) was derived from the following equation: C_t_ = C_∞_ (1 − e^−rt^) + C_0_, where C_t_, C_∞_, and C_0_ indicate the intracellular SO_4_^2−^ concentrations measured at time t, ∞, and 0, respectively, e represents the Neper number (2.7182818), r indicates the rate constant accounting for the process velocity, and t is the specific time at which the SO_4_^2−^ concentration was measured. The rate constant is the inverse of the time needed to reach ~63% of total SO_4_^2−^ intracellular concentration [82] and [SO_4_^2−^] L cells × 10^−2^ reported in figures represents SO_4_^2−^ micromolar concentration internalized by 5 mL RBCs suspended at 3% hematocrit.

#### 4.7.2. Experimental Conditions

Red blood cells (3% hematocrit), which were left untreated or treated with 20 mM H_2_O_2_ (1 h at 25 °C) with or without 10 µM Q (1 h at 37 °C before or after treatment with H_2_O_2_), were centrifuged (Neya 16R, 4 °C, 1200× *g*, 5 min) to replace the supernatant with SO_4_^2−^ medium. The rate constant of SO_4_^2−^ uptake was determined as described for the control condition.

### 4.8. Experimental Data and Statistics

All data are expressed as arithmetic mean ± standard error of the mean. For statistical analysis and graphics, GraphPad Prism (version 8.0, GraphPad Software, San Diego, CA, USA) and Excel (Version 2019, Microsoft, Redmond, WA, USA) software were used. Data normality was verified with the D’Agostino and Pearson Omnibus normality test. Significant differences between mean values were determined by one-way analysis of variance (ANOVA), followed by Student’s *t*-test, Bonferroni’s multiple comparison post-test, or ANOVA with Dunnet’s post-test, as appropriate. Statistically significant differences were assumed at *p* < 0.05; (*n*) corresponds to the number of separate measurements.

## 5. Conclusions

We conclude that exposure to 20 mM H_2_O_2_ induced OS in RBCs, which was manifested as morphological changes, as well as alteration of B3p and CD47 protein surface expression. Pre-treatment with Q avoided the formation of acanthocytes observed after exposure to 20 mM H_2_O_2_, prevented H_2_O_2_-induced OS damage, including ROS production, lipid peroxidation as well as protein oxidation, and restored the distribution of B3p and CD47 expression on the plasma membrane. Moreover, H_2_O_2_ exposure was associated with a reduction of the rate constant of SO_4_^2−^ uptake through B3p, as well as MetHb formation. Both alterations have been attenuated by pre-treatment with Q. Conversely, Q post-treatment prevented H_2_O_2_-induced OS damage, including ROS production, lipid peroxidation as well as protein oxidation, restored the distribution of B3p and CD47 on the plasma membrane, but only partially avoided the formation of acanthocytes. Importantly, post-treatment with Q did not restore the rate constant of SO_4_^2−^ uptake through B3p and did not inhibit the MetHb formation. These findings reveal that some, but not all, OS-related alterations are reversible, and that early application of antioxidants is envisioned to counteract the majority of OS-induced derangements. The present investigation provides mechanistic insights into the higher number of benefits deriving from the use of naturally occurring flavonoids against OS on a cellular level. The results obtained here confirm that measurement of B3p anion exchange capability remains a suitable tool for monitoring the impact of OS on RBC homeostasis. Considering the involvement of OS in a wide range of pathologies, new OS biomarkers, with both diagnostic and monitoring potential, are needed. Blood can be obtained from patients with minimally invasive procedures, reflects the physiological status of peripheral tissues, and therefore might represent a convenient source of OS biomarkers. We propose monitoring of B3p expression and function as a novel OS biomarker. In this light, future investigations are needed to clarify the signaling underlying the protective activity of Q on normal anion exchange ability, including possible effects on the interaction between B3p and the cytoskeletal proteins ankyrin and spectrin, their potential post-translation modifications, as well as a possible influence on the endogenous antioxidant machinery.

## Figures and Tables

**Figure 1 ijms-23-10991-f001:**
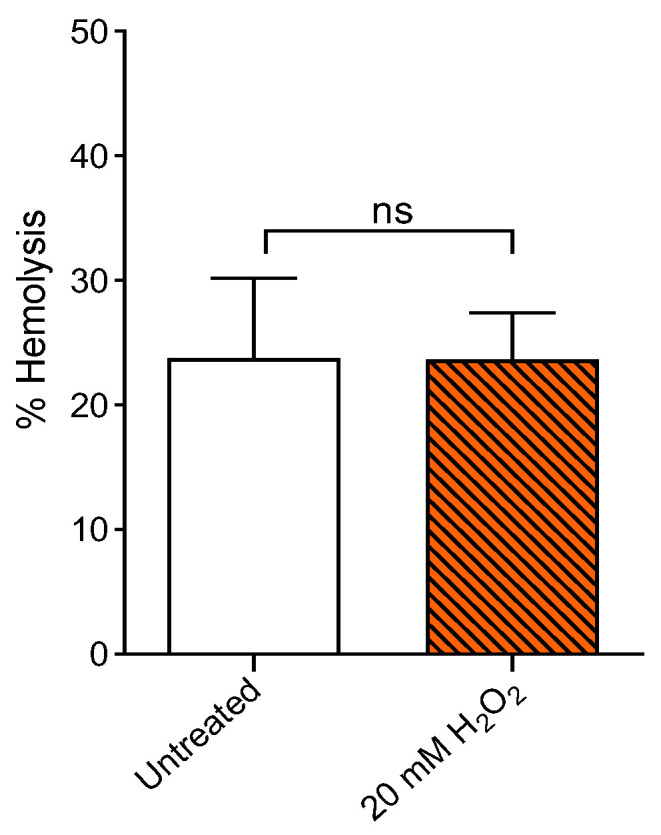
Hemolysis measurement. Percentage of hemolysis in RBCs left untreated (control) and in RBCs treated for 1 h (25 °C) with 20 mM H_2_O_2_. ns, not statistically significant versus control, paired Student’s *t*-test (*n* = 10).

**Figure 2 ijms-23-10991-f002:**
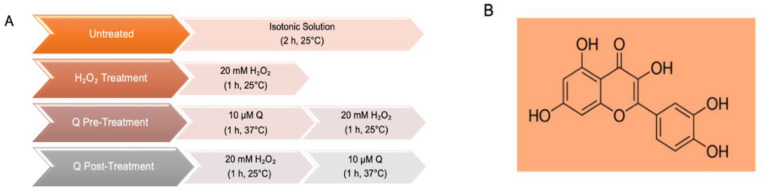
(**A**) Time course of experimental procedures. (**B**) Chemical structure of quercetin.

**Figure 3 ijms-23-10991-f003:**
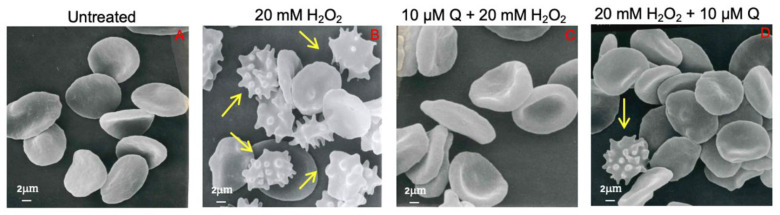
Erythrocyte morphology evaluation. Representative SEM images showing RBCs with a typical biconcave form ((**A**), left untreated, control); with surface blebs ((**B**,**D**) acanthocytes, arrows) after treatment with 20 mM H_2_O_2_ or 20 mM H_2_O_2_ + 10 µM Q (**C**). Pre-treatment with Q (10 µM) attenuated the morphological changes compared to H_2_O_2_ treatment. Vice versa, (**D**) post-treatment with Q (10 µM) still revealed a notable cell morphology alteration. Magnification 3000×.

**Figure 4 ijms-23-10991-f004:**
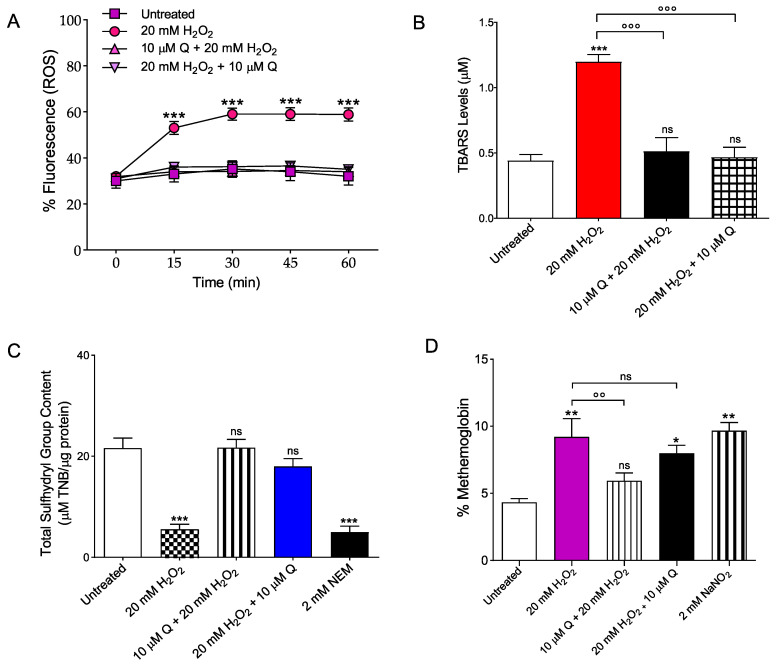
Determination of oxidative stress. (**A**) Detection of reactive oxygen species (ROS) levels by flow cytometry. Time course of ROS production in RBCs left untreated (control) or treated for 1 h at 25 °C with 20 mM H_2_O_2_ with or without pre-exposure to 10 µM Q for 1 h at 37 °C or pre-treated with 20 mM H_2_O_2_ and then exposed to 10 µM Q. ns, not statistically significant versus control; *** *p* < 0.001 versus control, one-way ANOVA followed by Bonferroni’s post-hoc test (*n* = 5). (**B**) Detection of TBARS levels. TBARS levels (µM) in RBCs left untreated (control) or treated for 1 h at 25 °C with 20 mM H_2_O_2_ with or without pre-incubation (1 h, 37 °C) with 10 µM Q or pre-treated with 20 mM H_2_O_2_ and then exposed to 10 µM Q. ns, not statistically significant versus control; *** *p* < 0.001 versus control; °°° *p* < 0.001 versus 20 mM H_2_O_2_, one-way ANOVA followed by Bonferroni’s post-hoc test (*n* = 10). (**C**) Sulfhydryl group content evaluation. Sulfhydryl group content (µM TNB/µg protein) in RBCs left untreated (control) and in RBCs treated for 1 h at 25 °C with 20 mM H_2_O_2_ with or without pre-exposure (1 h, 25 °C) to 10 µM Q or pre-treated with 20 mM H_2_O_2_ and then exposed to 10 µM Q. A 2 mM NEM was used as a positive control. ns, not statistically significant versus control; *** *p* < 0.001 versus control, ANOVA with Dunnet’s post-test (*n* = 10). (**D**) Methemoglobin (% MetHb) content. Red blood cells were left untreated or incubated with 20 mM H_2_O_2_ (1 h at 37 °C) with or without pre-exposure to 10 µM Q (1 h at 37 °C), or pre-treated with 20 mM H_2_O_2_ and then exposed to 10 µM Q. NaNO_2_ (2 mM for 1 h) was used as positive control. ns, not statistically significant; *,** *p* < 0.05 *p* < 0.01 versus control; °° *p* < 0.01 versus 20 mM H_2_O_2_, one way ANOVA followed by Bonferroni’s post-hoc test (*n* = 10).

**Figure 5 ijms-23-10991-f005:**
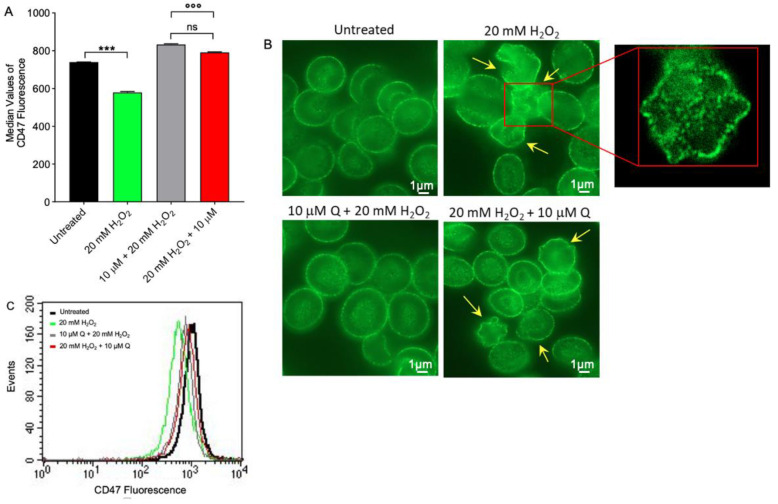
Flow cytometry analysis and imaging of CD47 protein expression. (**A**) Red blood cells were treated for 1 h with 20 mM H_2_O_2_ with or without preincubation for 1 h with 10 µM Q or pre-treated with 20 mM H_2_O_2_ and then exposed to 10 µM Q. Histograms represent median values of fluorescence intensity. In (**B**), representative images of CD47 expression obtained by immunofluorescence are shown (acanthocytes, arrows). Samples were observed with a 100× objective. Inset magnification, 7000×. In (**C**), typical flow cytometry measurements of CD47 expression of a representative experiment are shown. Note the significant morphological changes in H_2_O_2_ treatment as well as in 20 mM H_2_O_2_ + 10 µM Q. ns, not statistically significant; *** *p* < 0.001; °°° *p* < 0.001, one-way ANOVA followed by Bonferroni’s post-hoc test (*n* = 5).

**Figure 6 ijms-23-10991-f006:**
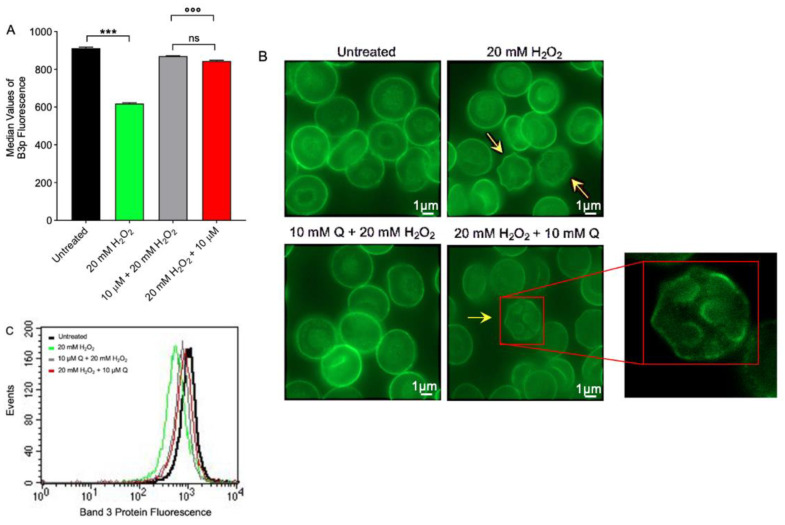
Flow cytometry analysis and imaging of B3p protein expression. (**A**) Red blood cells were treated for 1 h at 25 °C with 20 mM H_2_O_2_ with or without pre-incubation for 1 h at 37 °C with 10 µM Q or pre-treated with 20 mM H_2_O_2_ and then exposed to 10 µM Q. Histograms represent median values of fluorescence intensity. In (**B**), representative micrographs obtained by immunofluorescence showing B3p distribution in RBCs left untreated, treated with 20 mM H_2_O_2_, or alternatively, pre- or post-treated with 10 µM Q are shown (acanthocytes, arrows). Samples were observed with a 100× objective. Inset magnification, 7000×. In (**C**), typical flow cytometry measurements of B3p expression of a representative experiment are shown. Note the significant morphological modifications in H_2_O_2_ incubation as well as in 20 mM H_2_O_2_ + 10 µM Q. ns, not statistically significant; *** *p* < 0.001 versus left untreated (control); °°° *p* < 0.001, one-way ANOVA with Bonferroni’s multiple comparison post-hoc test (*n* = 5).

**Figure 7 ijms-23-10991-f007:**
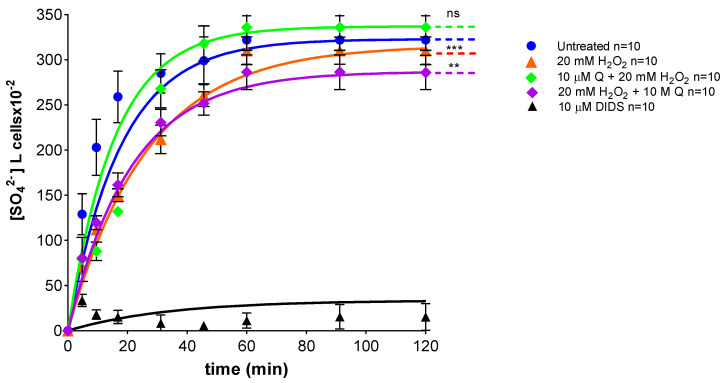
Time course of SO_4_^2−^ uptake. RBCs were left untreated (control) or treated with 20 mM H_2_O_2_ with or without pre-exposure to 10 µM Q for 1 h at 37 °C, or pre-treated with 20 mM H_2_O_2_ and then exposed to 10 µM Q, or exposed to 10 µM DIDS. ns, not statistically significant versus control; *** *p* < 0.001 versus control; ** *p* < 0.01 versus control, one way ANOVA followed by Bonferroni’s post-hoc test.

**Table 1 ijms-23-10991-t001:** Percentage of morphological alterations in RBCs left untreated (control) or treated as indicated. Data are presented as means ± S.E.M. from three independent experiments, where ns, not statistically significant versus control; *** *p* < 0.001 versus control; ** *p* < 0.01 versus control; ^^^^^ *p* < 0.001 versus 20 mM H_2_O_2_; °° *p* < 0.01 versus 20 mM H_2_O_2_ and control, one-way ANOVA followed by Bonferroni’s multiple comparison post-hoc test.

	Biconcave Shape	Acanthocytes
Control	93.8% ± 0.016	6.2% ± 0.011
20 mM H2O2	59.9% ± 0.010 ***	40.1% ± 0.010 ***
10 µM Q + 20 mM H2O2	90.4% ± 0.010 ^ns^	9.6% ± 0.010 ^ns, ^^^^
20 mM H2O2 + 10 µM Q	81.6% ± 0.009 ^ns^	18.4% ± 0.009 **^,^ °°

**Table 2 ijms-23-10991-t002:** Rate constant of SO_4_^2−^ uptake and amount of SO_4_^2−^ trapped in RBCs left untreated (control) and RBCs treated as indicated. Results are presented as means ± S.E.M. from separate (*n*) experiments, where ns, not statistically significant versus left untreated or 20 mM H_2_O_2_; ** *p* < 0.01; *** *p* < 0.001 versus control; °° *p* < 0.01 versus 20 mM H_2_O_2_, one-way ANOVA followed by Bonferroni’s multiple comparison post-hoc test.

Experimental Conditions	Rate Constant (min^−^^1^)	Time (min)	*n*	SO_4_^2−^ Amount Trapped after 45 min of Incubation in SO_4_^2−^ Medium [SO_4_^2−^] L Cells × 10^−2^
Control	0.056 ± 0.001	17.58	10	299 ± 18.63
10 µM Q	0.054 ± 0.001 ^ns^	17.55	10	298 ± 17.99 ^ns^
20 mM H_2_O_2_	0.042 ± 0.001 ***	26.33	10	266 ± 16.50 ***
10 µM Q + 20 mM H_2_O_2_	0.062 ± 0.001 ^ns^	15.99	10	286 ± 15.49 ^ns^
20 mM H_2_O_2_ + 10 µM Q	0.047 ± 0.001 **^,^°°	21.11	10	250 ± 19.80 **^,ns^
10 µM DIDS	0.017 ± 0.001 ***	61.50	10	5.49 ± 3.50 ***

## Data Availability

The data that support the findings of this study are available from the corresponding author upon reasonable request.

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
