# Peer review of "Antioxidant Activity of Quercetin in a H2O2-Induced Oxidative Stress Model in Red Blood Cells: Functional Role of Band 3 Protein"

_ijms, 2022, doi:10.3390/ijms231910991_

Round 1
Reviewer 1 Report
The Authors reported on the effect of pre- and post-treatment of quercetin on cellular oxidative stress using red blood cell erythrocytes as model.
The research is original and very well designed.
The description of the methods, as well as the presentation of the results are of high quality.
To the best of my knowledge this study is the first of this quality using this biological model, which is highly relevant to evaluate the in vivo effect of quercetin as antioxidant.
I strongly recommend this nice work for publication in IJMS.
I have only very minor suggestions about the figures and tables presentations:
Figure 3: add a,b,c, and d for the 4 pictures in order to ease their identification.
Figure 4: avoid cutting the figure from its legend.
Same remark for Table 2.
Author Response
- Point by point reply to the Reviewer 1
The Authors reported on the effect of pre- and post-treatment of quercetin on cellular oxidative stress using red blood cell erythrocytes as model. The research is original and very well designed. The description of the methods, as well as the presentation of the results are of high quality. To the best of my knowledge this study is the first of this quality using this biological model, which is highly relevant to evaluate the in vivo effect of quercetin as antioxidant. I strongly recommend this nice work for publication in IJMS. We thank the Reviewer for the overall positive evaluation.
I have only very minor suggestions about the figures and tables presentations:
Figure 3: add a, b, c, and d for the 4 pictures in order to ease their identification. We thank the reviewer for this suggestion. Done
Figure 4: avoid cutting the figure from its legend. We thank the reviewer for this suggestion. Done
Same remark for Table 2. We thank the reviewer for this suggestion. Done
Reviewer 2 Report
This study describes the potential mechanisms of reduction in oxidative stress-induced RBC injury by quercetin. The authors have described the changes related to Band 3 protein and CD47 as possible mechanisms in the protective effect of this compound. The experimental methods of the study are clearly described and the changes in erythrocytes are shown through various techniques. A few points need to be addressed:
- Throughout the paper there is inconsistent use of the terms RBC or erythrocytes. The authors must stick to either of the terms throughout the paper.
- The scale used in immunofluorescence images in Fig. 5 and 6 is not shown. Please provide the µm of measurement within the panels so readers can get an idea of RBC cell size changes.
- Expand DIDS in the methods and provide IC50 of this agent.
- The exact role of SO42- uptake via Band3 protein in human RBC is unclear and not well understood. The mechanisms previously reported should be explained and how measuring SO42- provides an idea about the Band3 activity. Is the activity reported in RBC of other species?
Author Response
- Point by point reply to the Reviewer 2
This study describes the potential mechanisms of reduction in oxidative stress-induced RBC injury by quercetin. The authors have described the changes related to Band 3 protein and CD47 as possible mechanisms in the protective effect of this compound. The experimental methods of the study are clearly described and the changes in erythrocytes are shown through various techniques. We thank the Reviewer for the overall positive evaluation.
A few points need to be addressed:
Throughout the paper there is inconsistent use of the terms RBC or erythrocytes. The authors must stick to either of the terms throughout the paper. We thank the reviewer for this suggestion. Done
The scale used in immunofluorescence images in Fig. 5 and 6 is not shown. Please provide the µm of measurement within the panels so readers can get an idea of RBC cell size changes. We thank the reviewer for this suggestion. Done
Expand DIDS in the methods and provide IC50 of this agent. We thank the reviewer for this suggestion. A reference has been added in the text.
The exact role of SO42- uptake via Band 3 protein in human RBC is unclear and not well understood. The mechanisms previously reported should be explained and how measuring SO42- provides an idea about the Band3 activity. Is the activity reported in RBC of other species? We want to thank the Reviewer for raising this point. In physiological conditions, Band 3 protein (B3p) favors chloride/bicarbonate (Cl-/HCO3-) electroneutral exchange across the plasma membrane. Since one of the peculiar features of erythrocytes is to rapidly exchange anions, it has been postulated that SO42- exchange may share a common mechanism with Cl- and HCO3- (Cabantchik et al., 1978), with the advantage that a turbidimetric method may more easily reveal the presence of transported SO42- than Cl-. In this regard, the rate constant for SO42- transport can be determined by a turbidimetric method, aimed at quantifying SO42- internalized through B3p as a function of time. Such in vitro technique, though not resembling the in vivo environment of erythrocytes, is a simple way to assess B3p anion exchange capability, as previously demonstrated by Jennings, 1976, Romano & Passow, 1984, Markovich, 2001, Romano et al., 2002, Teti et al., 2005 and Gugliotta et al., 2012.